# Migrant Men Living in Brazil during the Pandemic: A Qualitative Study

**DOI:** 10.3390/ijerph21010109

**Published:** 2024-01-18

**Authors:** Ramon Vinicius Peixoto da Silva Santos, João Cruz Neto, Sidiane Rodrigues Bacelo, Oscar Yovani Fabian José, Oscar Javier Vergara-Escobar, Felipe Machuca-Contreras, Maria Cecilia Leite de Moraes, Luciano Garcia Lourenção, Álvaro Francisco Lopes de Sousa, Layze Braz de Oliveira, Isabel Amélia Costa Mendes, Anderson Reis de Sousa

**Affiliations:** 1School of Nursing, Federal University of Bahia, Salvador 40110-909, BA, Brazil; ramon.vinicius@ufba.br (R.V.P.d.S.S.); leimo7@hotmail.com (M.C.L.d.M.); anderson.sousa@ufba.br (A.R.d.S.); 2Nursing Department, University for International Integration of the Afro-Brazilian Lusophony, Redenção 43900-000, CE, Brazil; joao.cruz@aluno.unilab.edu.br; 3Federal University of Rio Grande, Rio Grande 96203-900, RS, Brazil; dra.sidiane@outlook.com; 4Faculty of Nursing, Veracruz University, Minatitlán Campus, Veracruz 91700, Mexico; ofabian@uv.mx; 5Juan N Corpas University Foundation, Cra. 111 #159a-61, Bogotá 111321, Colombia; ojvergarae@unal.edu.co; 6Virrectoría de Investigación y Postgrado., Universidad Autónoma de Chile, Santiago 7500912, Chile; felipe.machuca@uautonoma.cl; 7Ministry of Social Security, Brasília 70059-900, DF, Brazil; luciano.lourencao@previdencia.gov.br; 8Institute of Teaching and Research Sírio Libanês Hospital, São Paulo 01308-050, SP, Brazil; 9Ribeirão Preto College of Nursing, Universidade de São Paulo, Ribeirão Preto 14040-902, SP, Brazil; layzebraz@usp.br (L.B.d.O.); iamendes@eerp.usp.br (I.A.C.M.)

**Keywords:** human migration, men’s health, Transcultural Nursing, COVID-19

## Abstract

This study aims to analyze the repercussions of the ongoing COVID-19 pandemic on the health of male immigrants, refugees, and asylum seekers in Brazil. A qualitative study involving 307 adult men living in Brazil during the COVID-19 pandemic was conducted. Data were collected between August 2021 and March 2022 and interpreted based on the Transcultural Nursing Theory. Cultural care repercussions were identified in various dimensions: technological: changes in daily life and disruptions in routine; religious, philosophical, social, and cultural values: changes stemming from disrupted social bonds, religious practices, and sociocultural isolation; political: experiences of political partisanship, conflicts, government mismanagement, a lack of immigration policies, human rights violations, and xenophobia; educational/economic: challenges arising from economic impoverishment, economic insecurity, unemployment, language difficulties, and challenges in academic and literacy development during the pandemic. The persistence of the COVID-19 pandemic in Brazil had significant repercussions for the health of migrant men, resulting in a transcultural phenomenon that requires sensitive nursing care. Implications for nursing: the uniqueness of cultural care in nursing and health, as most of the repercussions found were mostly negative, contributed to the increase in social and health vulnerabilities.

## 1. Introduction

Individuals undergoing physical, environmental/territorial, and sociocultural changes due to their transition from their current living spaces to other locations face numerous challenges. These transitions often occur for reasons such as seeking economic opportunities, fleeing wars, natural disasters, and diseases like COVID-19 [1]. Communicable diseases pose a high risk to migrant populations, particularly given their vulnerabilities [2,3]. Recognizing the need for global efforts to support the migrant population during the pandemic, the Pan American Health Organization (PAHO) recommended non-pharmacological measures to combat the pandemic. Additionally, the United Nations (UN), through the United Nations High Commissioner for Refugees (UNHCR), called for the protection of migrant populations in this context [4].

Migration in Brazil has led to detrimental repercussions for life and health, considering the persistence of the disease, weaknesses in government management, delays in vaccination, and the establishment of sanitary measures for this population group [5]. The literature already observes deleterious impacts on physical, psychosocial, spiritual health, transcultural cohesion, religious, familial/affectionate dimensions, socioeconomic conditions, politics, and education, especially among the male population [6,7,8].

In addition to the epidemiological control of COVID-19 through the reduction in undocumented immigration across unprotected natural borders [9], other global health emergencies must be considered among male migrant groups. These include seeking refuge and political asylum, conflicts and violence, xenophobia, racism, impoverishment, hunger, misery, food insecurity, economic and employment fragility, abusive alcohol and drug use, immunosuppression, homelessness, severe mental suffering, and involvement in criminal activities such as drug trafficking, as these issues impact the social and psychological well-being, quality of life, and health promotion of men worldwide [10,11].

In health crisis situations affecting migrant populations, accessing culturally adapted knowledge is essential, as advised by the Transcultural Nursing Theory. This theory aims to achieve cultural care production, considering cultural practices consistent with the values and beliefs of the populations being assisted in their environments [6]. This theoretical framework can contribute to specialized healthcare, with qualified nursing and healthcare professionals addressing patients’ unique demands and needs [8,12,13,14].

Studies conducted in other countries have already pointed to deficiencies in healthcare and the presence of social health inequalities among the male migrant population, exposing barriers to accessing healthcare services, vulnerabilities in mental health, and fragilities in the social support network [15,16,17,18].

Considering the lack of studies investigating the pandemic’s repercussions on the lives and health of men in a transcultural context, this study aimed to fill this gap and provide significant findings regarding its impacts nursing and healthcare practice. Therefore, this study was guided by this research question: how may the persistence of the COVID-19 pandemic in Brazil have affected the health of immigrant, refugee, and asylum-seeking men? The objective of this study was to analyze the repercussions of the persistence of the COVID-19 pandemic on the health of immigrant, refugee, and asylum-seeking men in Brazil.

## 2. Materials and Methods

### 2.1. Study Type

This qualitative, descriptive, and multicenter study was conducted online, covering all regions of Brazil.

### 2.2. Theoretical–Conceptual Framework

This study is grounded in the Transcultural Care Theory proposed by Leininger (1985) [6]. This theory provides a framework for analyzing the cultural and social patterns of individuals, identifying how they are related to their environment and culture. It encompasses concepts such as social structure, culture, care, cultural values, and the diversity and universality of care. This allows for us to understand how the elements of society influence an individual’s health [6].

### 2.3. Location

This study was conducted in a hybrid environment (in-person and virtual). The physical environment included places such as hotels/hostels, street markets, theater arenas, circuses, beach stalls, language schools, embassies, and local businesses. In total, 40 locations in 12 municipalities were visited. The virtual environment included digital social media platforms such as Instagram, Facebook, WhatsApp, and Telegram, as well as email communications. For this purpose, 125 institutions/organizations/departments providing social support to migrant populations were mapped by a trained research team composed of a graduate student, a master’s student, a doctorate holder, and three nursing doctors involved in teaching and research at the time of data collection, who had no direct connections to the investigated participants.

### 2.4. Participants

This study included cisgender and/or transgender adult male migrants (international immigrants, refugees, and asylum seekers, whether forced or voluntary) living in Brazil. Stateless men were not included in this study. Considering the scarcity of studies of migration, particularly with regard to men, the decision was made to focus on a sample of cisgender and transgender men due to the specificity of this population. This group has not yet been discussed in the literature in the context of the pandemic, which places them in a position of vulnerability compared to other populations and justifies their inclusion.

### 2.5. Data Collection

Data were collected between August 2021 and March 2022 through two methods. The first, asynchronous method involved participants contacted through digital social media. The second, synchronous method involved face-to-face interactions between participants and the research team in a single meeting, with an average duration between 20 min (self-administered virtually) and 40 min (in-person). In both strategies, a semi-structured questionnaire hosted on the Google Forms^®^ platform [19] was used (Google, Mountain View, CA, USA). The COREQ (COnsolidated criteria for REporting Qualitative research) Checklist was used and included a Visual Informed Consent Form and structured questions about personal characteristics, employment, health status, migration, pandemic repercussions, and social vulnerabilities. It also included open-ended questions such as the following:Tell us about your daily life during the COVID-19 pandemic;Tell us about your health during the pandemic;Have you experienced any problems? How have you dealt with or faced these problems?;Do you believe being a migrant has caused any harm to your health during the pandemic?;Did you receive any assistance to deal with the harm caused?

The snowball recruitment technique was used for participant selection. Initially, the first 10 participants sampled were called “seeds”. They were encouraged to invite new participants (a “one refers another” strategy), resulting in 17 “children”, one from each Brazilian state, joining this study [16]. This followed the criteria of theoretical sampling [20,21]: convergences, complementarities, occurrences, and theoretical data density, resulting in a sample of 307 participants.

### 2.6. Data Analysis

The collected data underwent verification to ensure data validity and reliability, including checking for duplicate, incomplete, and/or inconsistent data. Data were extracted from the Excel^®^ spreadsheet provided by the Google Forms^®^ platform and compiled into internal databases. They were stored in designated folders with specific codes and identification and maintained on institutional devices for manipulation by authorized personnel. Subsequently, the data were coded, the corpus was organized for analysis, and processing was carried out using the IRAMUTEQ software (Psycom, Toulouse, France). This allowed for similarity analysis and the generation of an analytical tree, in addition to thematic reflexive content analysis, which derived a major thematic category consisting of four subcategories [21]. Finally, the data were interpreted based on the Transcultural Care Theory proposed by Leininger (1985) [6].

### 2.7. Ethical Aspects

This study adhered to ethical aspects outlined in Resolution 466/2012 concerning research involving human subjects. This project received approval from the Research Ethics Committee and complied with current Brazilian legislation. Only participants who signed the informed consent form were eligible for participation in this study. To ensure participant anonymity, pseudonyms were used: H for man, number 01 for order of collection, and the country of origin, for example, H01—China.

## 3. Results

The findings of this study are presented based on participant characteristics and the description of empirical data, organized into categories and subcategories, deduced according to the assumptions of the adopted theoretical framework.

Who Are We Talking About? Where Are the Participants From? Characterization of the Study’s Target Audience.

This study included male immigrants, refugees, and asylum seekers from seven continents. A total of 307 participants were included, ranging in age from 18 to 76 years old. Among them, 94.8% identified as cisgender, 81.4% as heterosexual, 11.7% as homosexual, 3.6% as bisexual, 42.7% as White, 29.3% as Black, 17.6% as Mixed Race, 6.8% as Indigenous, and 3.6% as Asian. Furthermore, 25.1% had completed higher education, 66.1% resided in states in the Northeast region, 70% lived in rented houses or apartments, 50.2% had permanent residence status, 33.2% were self-employed, and 86% had migrated to Brazil before the COVID-19 pandemic.

Participants included men from 7 geographical regions: Africa (83), Central America (20), North America (8), South America (110), Asia (34), Europe (50), and Oceania (2). In terms of nationalities, 65 different countries were represented: Germany (6), Argentina (29), Angola (14), Algeria (1), Australia (2), Belgium (1), Benin (5), Bolivia (4), Cape Verde (10), Canada (2), Chile (13), China (16), Colombia (14), the Ivory Coast (1), Costa Rica (5), Cuba (1), Croatia (1), Egypt (1), El Salvador (3), Ecuador (4), Spain (2), USA (3), Finland (1), France (8), Ghana (4), Guangdong (1), Guatemala (1), Guinea-Bissau (17), Greece (1), Haiti (5), Honduras (2), India (2), England (4), Iran (1), Ireland (3), Israel (2), Italy (12), Japan (2), Mali (1), Malaysia (1), Morocco (1), Mexico (3), Mozambique (11), Nigeria (4), Norway (1), Palestine (2), Pakistan (1), Paraguay (4), Peru (19), Portugal (4), the Dominican Republic (3), the Republic of the Congo (1), Russia (1), São Tomé and Príncipe (3), Senegal (7), Serbia (1), Syria (3), Switzerland (3), Taiwan (1), Togo (1), Turkey (1), Ukraine (1), Uruguay (8), Uzbekistan (1), and Venezuela (15).

### 3.1. Empirical Result

The similarity analysis revealed central words in the participants’ discourse. The most frequently occurring terms included “stay” (300 times), “deal with” (263 times), “experience” (238 times), and “bring” (222 times), as well as “seek refuge” (157 times), “seek asylum” (153 times), “go through” (147 times), “work” (124 times), and “feel” (118 times). Participants’ statements, such as “Having to stay without work affected my sense of life”, “My daily life during the COVID-19 pandemic had a significant and overall impact on my life, especially socially because I had to deal with family problems”, “My daily life during the COVID-19 pandemic was very challenging (…) I started to experience the virtual”, and “I felt like someone who could bring danger to others, which makes me vulnerable”, were validated through this analysis.

### 3.2. Thematic Category 01: Repercussions on Health and Care

This thematic category encompasses four subcategories that indicate the constituent elements of Transcultural Care, namely technological factors; religious, philosophical, social, and cultural factors; political and legal factors; and educational and economic factors.

### 3.3. Thematic Subcategory 1A: Technological Factors—Transformations and Ruptures Leave Marks in Daily Life

Technological factors have generated transformations and disruptions in the lives of migrants due to the pandemic and its persistence in Brazil. Restricted access to the healthcare system (as a healthcare technology), as well as to goods and services in general, changed the landscape of self-care. Migrant men, regardless of their region of origin, experienced repercussions from restrictive, prolonged, and unpredictable health measures.

In response to the need for isolation, time optimization allowed for the acquisition of new job skills and the adoption of new sanitary habits in daily life, social and domestic interactions, and in the workplace (home office, telecommuting, or delivery). Consequently, due to the high number of COVID-19 cases in Brazil, a detrimental consequence for the lives of immigrant men in this country was having to experience the sequelae of the disease, as evidenced by the following statements:Africa: “Many changes in daily life, with restrictions on leisure, physical activity, and classes, which shifted to distance learning” (H31—Cape Verde); “The workload increased.” (H166—Cape Verde)Americas: “I had problems with contacting my family; we were separated.” (H51—Cuba); “Work became solitary.” (H139—Canada); “We spent many days confined, a complete change in habits and routines, in the face of the uncertainties of the pandemic in Brazil.” (H258—USA); “I had to stay at home and adapt to working from home.” (H03—Argentina); “I had to plan how to survive in the country.” (H122—Argentina).Asia: “I work on the street; with businesses closed, I had to transform my way of surviving. I suffered a lot.” (H59—China); “There were days that seemed to never end.” (H83—Pakistan); “It completely affected a healthy lifestyle.” (H262—Malaysia).Europe and Oceania: “It was difficult to adapt to working, as it became intense and exhausting.” (H107—Spain); “It was chaotic; the changes prevented me from accessing public leisure spaces, study, and work.” (H274—Greece); “I had to start with delivery work.” (H103—Australia); “New challenges emerged, such as having to live with sequelae caused by COVID-19.” (H253—Australia).

### 3.4. Thematic Subcategory 1B: Religious, Philosophical, Social, and Cultural Values—Being Distant from the Country of Origin Brings Socio-Cultural Disintegration

Perceived as a difficult, complex, concerning, and detrimental situation, the persistence of the pandemic had repercussions that impacted men’s health biopsychosocially and spiritually. The reflexive thematic content highlighted the challenge of being a father during the pandemic; marital conflicts; reduced interaction with affective–conjugal partners due to an increased workload in the home office; distancing from physical contact, social and affective interaction, and religious practices with significant people in their cultural network; disconnection from one’s inner self; and anxiety, sadness, decreased empathy, loneliness, and grief due to the loss of family and friends affected by COVID-19.

The reduction in the number of cultural encounters with members of the communities from their countries of origin had adverse effects on maintaining bonds and cultural interaction. Men attempted to strengthen their family ties, optimize the use of the home environment, and engage in religious practices, as exemplified below:Africa: “I had to deal with the unexpected birth of a child during a difficult, stressful, and uncertain period, relying on people only virtually since we were isolated at home, fearing that we could be infected by the virus.” (H49—Nigeria); “I felt the impacts on my connection with my inner self.” (H201—Senegal).Americas: “I had to deal daily with difficulties and internal struggles.” (H194—Costa Rica); “Being far from my family, the fear of getting sick from COVID-19, and the fact that I was distant from the culture of my home country left me stressed, anxious, and worried.” (H264—The Dominican Republic); “I went through anxious days with the situation, but I sought God and prayed.” (H224—Mexico); “I lost many family members and friends to COVID-19 in my home country, and unfortunately, I couldn’t be with them to grieve and say goodbye.” (H04—Venezuela); “I had to face depression, affecting my mental health due to constant fear.” (H12—Argentina).Asia: “They were days of social insecurity.” (H130—Israel); “I was unable to have leisure time with my family or travel to my home country, leaving me with a very bad feeling.” (H151—Syria); “A very bad feeling, not being able to have meetings and prayers with people from my country.” (H243—Iran).Europe and Oceania: “Everything became virtual, making me sad and lacking in empathy.” (H74—Germany); “Fear harmed the maintenance of the cultural ties I had.” (H77—Italy).

### 3.5. Thematic Subcategory 1C: Political and Legal Factors—Experiencing Governmental Crisis in Pandemic Management Generates Disbelief and Barriers in Coping

Political instability resulted in repercussions for psychological well-being, especially due to the government’s management of the pandemic. This context triggered negative emotions such as anger, confusion, and disbelief about the decisions of national, state, and municipal authorities, exposing individuals to socioeconomic and labor insecurity.

Africa: “I experienced a lot of apprehension, anxiety, and anger because many people did not follow the recommendations of health authorities, including the federal government.” (H32—Cape Verde).Americas: “I avoided looking at the news because I would cry and panic, noticing the lack of logistics, management, and care from the Ministry of Health for the people in Brazil. I was overwhelmed by such a political problem in the country.” (H12—Argentina); “They spent a lot on ineffective tests that gave errors. It was mismanagement, and this had significant impacts.” (H224—Mexico);Asia: “It was very stressful to deal with constant changes and uncertain government decisions.” (H48—Taiwan); “There was a lack of structure and political organization to deal with the pandemic. I became desperate.” (H181—Egypt); “I didn’t receive any support from the government. The rulers don’t think about immigrants. I had to work on the street, risking getting sick, without the right to quarantine.” (H195—Palestine).Europe and Oceania: “In Brazil, there were many disagreements in recommendations, even coming from the Brazilian government. There was a lot of political-party polarization.” (H103—Australia).

### 3.6. Thematic Subcategory 1D: Educational and Economic Factors—Prolonged Pandemic Experience Resulted in Economic and Educational Impoverishment

The persistence of the pandemic exacerbated the economic and educational vulnerabilities experienced by migrant men. Precarious working conditions, a lack of income, and difficulties accessing emergency public assistance were observed. This led to impoverishment, food and financial insecurity, as well as barriers related to language and delayed and interrupted studies.

Africa: “The cost of living became more expensive, and income decreased.” (H44—Senegal); “I experienced need, like hunger.” (H45—Senegal); “I couldn’t earn income. I faced financial difficulties.” (H61—Nigeria).Americas: “I became impoverished.” (H51—Cuba); “I didn’t even know how to speak Portuguese, which caused many difficulties in interaction.” (H228—Honduras); “The return to school was very late and harmed me.” (H245—El Salvador); “It was frustrating, with the routine changing with the children at home all the time.” (H204—Canada); “I was laid off from my job and had many bills to pay. I needed to support my family, and I managed to get food for my children, but it was desperate. I became paranoid.” (H02—Peru); “I had to look for food and shelter.” (H122—Argentina); “I became unemployed. It was a very difficult time, especially because I couldn’t rely on the Brazilian government’s emergency assistance.” (H211—Venezuela).Asia: “I had to close my business.” (H58—China); “The biggest impact was the lack of income to survive during confinement.” (H165—China); “I had to do online sales.” (H195—Palestine).Europe and Oceania: “I couldn’t sustain myself working in Brazil and had to return to France.” (H43—France); “My children stopped studying, and I, who am an English teacher, lost all my classes at the beginning of the pandemic.” (H256—Ireland); “In Brazil, there were many fake news spread about the disease.” (H253—Australia).

The COVID-19 pandemic resulted in repercussions for the cultural care of migrant men living in Brazil in this context. This phenomenon mobilized different transcultural factors of the male experience: technological, religious, social, philosophical/cultural, political, economic, and educational factors, as well as the elements that constitute them, which influenced the health and psychosocial well-being of men, as shown in the Sunrise Model (adapted), proposed by Madeleine Leininger, which is explained in Figure 1.

## 4. Discussion

This study investigated the repercussions of the persistence of the COVID-19 pandemic of men residing in Brazil, whether for immigration, refuge, or asylum purposes.

It should be noted that the transformations and disruptions in life experienced by migrants can have dimensions different from those who are not affected by this condition [10,11]. In this sense, our findings point to the uniqueness of cultural care in nursing and health, as most of the repercussions found were mostly negative, contributing to the increase in social and health vulnerabilities and potentially resulting in declines in the congruence and transcultural adaptation of individuals and healthcare teams.

According to the postulates of Transcultural Theory, this dimension of human care will occur in daily professional practices when assistance, support, facilitation, and/or empowerment actions and decisions are focused on the individual or their group or community to support optimized, beneficial, and satisfactory health promotion [6]. This is justified by the presence of assistance challenges faced by population groups experiencing international migration in Latin American countries, such as Chile, which still experiences technical and administrative difficulties, cultural barriers, and weaknesses in regulations and healthcare strategies, especially in the context of Primary Healthcare (PHC). This is compounded by the absence of records that accurately map the profile and quantity of the migrant population in that country and the lack of tools for the provision of culturally sensitive care [22].

Therefore, considering that the pandemic caused disruptions to everyday life, leaving harmful marks on the life and health of migrant men, it is recommended to invest in health technologies that value migrant communities in each country. This is important as the precariousness of social identity, the deficit in self-care capacity, and cultural transformations can perpetuate in the post-pandemic era and result in social malaise, such as the intensification of xenophobia and conflicts between peoples.

Not only in Latin American countries have the health needs of the migrant population, which also experiences refuge, been affected by health inequalities and inequities. In high-income countries such as Canada, England, and Italy, aspects like communication, continuity of care, and trust have been shown to influence the overcoming of legal, financial, geographical, and cultural issues, as evidenced in a systematic review [23].

With the presence of weakened familial ties, men may face difficulties in making decisions for maintaining good health and well-being due to the reduction in their socio-affective contact network and even face problems such as loneliness, low self-esteem, and unhappiness due to the impossibility of sharing difficulties and opportunities experienced in daily life. In this sense, interventions focused on promoting harmony and strengthening familial bonds should be reinforced, along with the expansion of humanitarian actions, the promotion of support groups, and the encouragement to create migrant communities to improve living conditions in their new countries of residence.

Furthermore, another factor of interest in the field of nursing and health is related to family breakdown, which, in the context of the pandemic, has become even more complex for migrant men who witness their partners’ gestational processes and experience fatherhood during periods of confinement. This exposes them to high levels of stress and a lack of social support in their socio-affective network, including healthcare services such as PHC, which were largely closed during the peak of the pandemic.

For the migrant population, the imbalance of cultural congruence with the family can result in psychosocial well-being and reproductive health issues, as confirmed by a study conducted in the Southern region of Brazil, which found late prenatal care initiation and delayed healthcare-seeking behavior among men [24,25]. It was observed during the pandemic that certain social groups were severely disadvantaged, with a greater emphasis on those considered at the base of the socioeconomic pyramid [26].

In light of findings that point to the separation of migrant men from their culture, religion, and social, affective, and marital relationships during the pandemic, there is a concern that without necessary support, these men may continue to face such challenges. This could result in worse health outcomes that are detriment to social well-being; impoverishment; widening inequalities; an increased risk of violence; including intrafamilial violence; and higher costs for health services.

In the male population, it is observed that the pandemic has generated conflicts in the regulation of emotions, leading to emotional suppression, a sense of threat due to the loss of the provider role, increased reactivity, weakened masculinity due to concerns about virility, and a decline in self-care and invulnerability [8,27]. These factors are compounded by issues related to socioeconomic status, race/ethnicity, stress, and intolerance [28]. Furthermore, a significant portion of the men who participated in this study reported migrating to Brazil before the context of the COVID-19 pandemic. In line with this, this study confirms, for example, the worsening of social vulnerability among Venezuelan migrants in Brazil [29].

The pandemic also had considerable psychological impacts, especially for those who are more vulnerable [30]. In the field of mental health, stress is a concerning factor in men, and this relationship can be mediated by the uncertainty of the health–disease process, which increases significantly when the population in question is composed of immigrants [31]. In this case, there is also an increased risk of mortality in this population, as well as subsequent growth in severity, regardless of age or hormonal factors [31]. Furthermore, the male population has a higher rate of intensive care unit admission, hospitalization, and progressive evolution to death compared to females [32].

It is strongly recommended that mental health be understood as a universal health right. This involves establishing successful programs that reduce stigma and discrimination related to male mental health and migration. Additionally, it is crucial to create new strategies for promoting mental health that encourage greater adherence among the male population.

Therefore, the way that men deal with the prevention and control of health issues, especially COVID-19, is linked to their health habits and susceptibility to diseases. This population also shows limited knowledge regarding healthcare [33]. Due to experiences of death and grief and weaknesses in their behavioral barriers, men are more susceptible to negative health phenomena such as COVID-19. In this context, it is essential to understand the dimension of this health issue in specific populations, focusing on the repercussions and impressions related to self-care and health promotion strategies [9,28].

It Is recommended to strengthen interventions for groups of men that consider and value their cultural specificities. These interventions should include strategies to ensure inter- and transcultural congruence, strengthen bonds, and build cultural support networks. This involves providing social support, acquiring a set of culturally structured behaviors that are competent in terms of facing complex situations such as environmental barriers, racism, stigma, xenophobia, and other forms of discrimination [34].

It is envisioned that without the appreciation of cultural care, migrant men may place less value on their own healthcare due to faced barriers, as well as the scarcity of resources for health promotion. It is recommended to invest in popular health actions that take into account the different cultures, languages, specificities, and needs of each community.

Considering that political and governmental factors have had negative impacts on the health of migrant men, it is imagined that without international public policies that address this population, the perception of a lack of assistance could further deter men from seeking essential health interventions, particularly in the pandemic and post-pandemic context.

Furthermore, supporting men in self-care activities and improving health and psychosocial well-being in their communities, especially among refugee and asylum-seeking men, is crucial. Additionally, strengthening nursing and health education in the context of migration and Transcultural Care is necessary, considering COVID-19 as a psychosociological phenomenon with significant implications for nursing knowledge and practice.

The impacts of COVID-19 are still being studied in all fields of health. However, the selection of strategies to mitigate the consequences of this health issue should be based on the characteristics of populations and symbolic values, especially in minority groups, such as migrant men. In this intersection, behavioral phenomena will be important means of seeking protection and epidemiological indicators [35].

It is known that when public or private organizations stimulate a contextualized view of health problems, relativizing individualism and enhancing the needs of a particular audience, institutional contributions begin to value redefined approaches based on gender intersections. This reinforces the need for initiatives that consider migrant populations from an intersectional perspective.

### 4.1. Implications for Health Policy

The implications for health policy stemming from the COVID-19 pandemic’s impact on immigrant, refugee, and asylum-seeking men in Brazil, as framed by Transcultural Theory, are profound. First and foremost, this study underscores the pressing need for culturally sensitive healthcare practices. Nurses and healthcare policymakers must recognize the unique challenges faced by these marginalized populations, such as language barriers, differing health beliefs, and limited access to healthcare services. This demands the development of targeted interventions that promote health equity, including the provision of culturally competent care and health education tailored to the specific needs of these communities. Additionally, health policies should prioritize addressing social determinants of health, such as economic disparities and housing instability, which disproportionately affect immigrant and refugee populations, as they play crucial roles in shaping health outcomes.

Furthermore, the pandemic’s effects on these vulnerable groups highlight the importance of a comprehensive and inclusive healthcare system. Policymakers must work towards removing barriers to healthcare access, including legal and bureaucratic hurdles that often prevent immigrants and refugees from seeking timely medical care. Additionally, the pandemic has exposed the need for improved data collection and surveillance methods to effectively monitor the health statuses of these populations. Nursing professionals can contribute by advocating for policies that promote equitable access to healthcare, offering training in cultural competence, and actively engaging with community organizations to bridge the gap between these underserved populations and the healthcare system. Ultimately, these implications underscore the vital roles of nurses and health policymakers in fostering health equity and social inclusion among immigrant, refugee, and asylum-seeking men in Brazil and similar settings worldwide.

This study’s findings hold relevance for the international nursing community, offering valuable insights into Transcultural Nursing and the impact of COVID-19 on migrant populations, with the potential to inform nursing practices and global healthcare policies.

### 4.2. Limitations and Strengths

This study has strong points, such as the identification of a vulnerable population with multiple transcultural needs. As a result, it serves as an inducer of public support policies for immigrants while highlighting the health needs of male immigrants during the pandemic. However, it has weaknesses, including the inability to attest to behavioral changes in other nationalities due to time constraints or their access to research.

This study has limitations. Although this study reached participants from all regions of Brazil, the results may reflect specific realities, which could reduce their generalizability. This study did not conduct a comparative analysis between men and women, which limits understanding of the phenomenon across genders. The recruitment strategy for participants may also have limited the study’s reach among different groups of men. The cultural and territorial particularities of each region in Brazil might have resulted in both specific experiences for each migrant man and difficulties in deepening the findings.

This article contributes to understanding the multifaceted challenges faced by male immigrants during the COVID-19 pandemic in Brazil. The use of the Transcultural Nursing Theory as an analytical framework offers a comprehensive lens to explore the cultural care implications arising from the pandemic. However, due to the small sample size, generalizations for the entire country of Brazil are not feasible. It was not possible to have a Brazilian control group due to the course of the pandemic and the changes imposed by immigration in this scenario.

## 5. Conclusions

The persistence of the COVID-19 pandemic in Brazil has significantly impacted the health of migrant men, evolving into a transcultural phenomenon requiring sensitive nursing care.

The affected transcultural factors include the following:

Technological factors: Transformations in daily life and disruptions caused by limited access to healthcare, goods, and services have led to changes in self-care practices and work habits. These changes encompass the adoption of new jobs, teleworking, and reliance on delivery services.

Religious, philosophical, social, and cultural values: the breakdown of family ties, distancing from emotional connections, changes in religious practices, and sociocultural isolation have led to a loss of transcultural cohesion.

Political factors: migrant men have been affected by political conflicts, partisan polarization, poor government management, an absence of immigration policies, violations of human rights, and xenophobia.

Educational and economic factors: These areas have been influenced by economic impoverishment, financial insecurity, unemployment, language barriers, and difficulties in social and emotional interactions. Additionally, there have been disruptions in education and academic attainment during the pandemic.

In response to these challenges, Transcultural Care has addressed the cultural specificities and demographic transformations resulting from the pandemic, focusing on the male experience.

## Figures and Tables

**Figure 1 ijerph-21-00109-f001:**
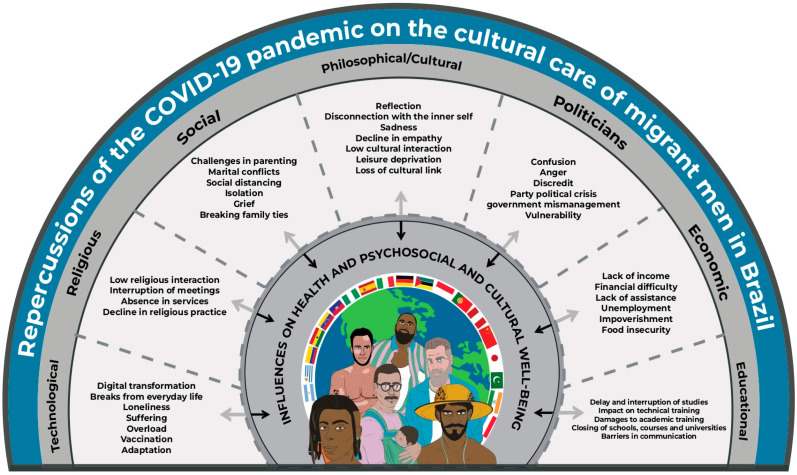
Explanatory Model of the Impact of the COVID-19 Pandemic on the Lives and Health of Immigrant, Refugee, and Asylum-Seeking Men in Brazil, Based on the Framework of Transcultural Theory.

## Data Availability

Data is contained within the article.

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
