# Peer review of "Migrant Men Living in Brazil during the Pandemic: A Qualitative Study"

_ijerph, 2024, doi:10.3390/ijerph21010109_

Round 1
Reviewer 1 Report
Comments and Suggestions for Authors
This qualitative study provides insight into the problems of health of immigrant, refugee, and asylum-seeking men in Brazil during the COVID-19 pandemic. It is well-written with concise conclusions, which are important for understanding the problems of the mentioned minority group. Excellent work! The only thing missing is the strengths and limitations of the conducted study. I would suggest that this article should be accepted for publication after adding these minor changes.
Author Response
Dear Reviewer,
Thank you for your insightful feedback on our qualitative study, which explores the health issues of immigrant, refugee, and asylum-seeking men in Brazil during the COVID-19 pandemic. We are pleased to hear that you found our study well-written, with concise conclusions that effectively highlight the challenges faced by this minority group. Your positive evaluation and recognition of our work as 'excellent' are greatly appreciated.
We acknowledge your suggestion regarding the inclusion of the strengths and limitations of our study. Understanding the importance of this aspect for a comprehensive view of our research, we agree that incorporating these details will further enhance the quality and depth of our article.
In response to your recommendation, we will include the necessary information on the strengths and limitations in the designated sections (Lines 471 to 492). This addition will provide a more balanced perspective and contribute to the overall transparency and validity of our findings.
We believe that these amendments will address your concern and make our paper more robust and informative for our readers. We are committed to making the necessary revisions and thank you for your constructive feedback. We look forward to the opportunity to present our revised manuscript for your review and hope for its acceptance for publication.
Reviewer 2 Report
Comments and Suggestions for Authors
Dear authors, I congratulate you on the scientific paper you have written. The topic is original, so far insufficiently examined in many articles about the COVID pandemic, and yet very important, especially for the so-called Western world, where migrants from all over the world are flocking from third countries. I believe that this aspect of concern for the health and well-being of migrants is less taken into account in the policies of governments towards them, and it is certainly significant in situations of public health crises such as a pandemic.
The research itself is methodologically well established. Clear results follow from the collected data and their processing, which provide answers to the hypotheses and goals of this article.
The conclusions are clear, precise, unambiguous and significant for wider social communities facing immigration crises.
However, there are minor errors and a couple of clarifications that I think will improve the quality of the work.
1. As for the title of the article itself, I think it should be expanded, because the very first word Health, according to me, does not include all aspects of life that are further explored in the research.
2. There is a typographical error at the beginning of line 21. There is an extra letter A.
3. I believe that keywords must be arranged according to the MeSH system, and that they must be a little more precisely related to the very essence of the work.
4. Literature citations throughout the text are not listed according to the rules. For example. in lines 56 and 71, in parentheses, it should read like this: 6-8 and 8, 12-4. This should be corrected throughout the work.
5. Regarding materials and methods and the Type of Study section
Should the type of study be specified more clearly in line 82 - descriptive, analytical, prospective, retrospective?
6. Regarding the section on study participants in line 97-100, I think it should be expanded. Provide clear inclusion and exclusion criteria in the examined group. It is also necessary to answer the question why only the male population, which is either cis or transgender, was selected for the research group, and why is it important and related to the research itself?
7. In line 122, a reference should be added that talks about the theoretical criteria for sampling, which are listed further on in the text.
8. There is a typographical error in line 136. A full stop is missing at the end of the sentence.
9. In line 153, I think the term "gay" does not fit, especially since the term heterosexual is in front of it. Please correct.
10. I also find the Results section lacking. There is no introductory graphic and textual sociodemographic and cultural presentation of the examined sample. That's really what the study lacks. Especially since further on in the results, typical responses of immigrants are presented in relation to their home region of origin. It is necessary to know how many respondents we have for each specific category of immigrants and what their structure is. It will be interesting to supplement the discussion with those data.
11. In the discussion in line 298, the word Health is also not sufficient in my opinion to describe all those aspects of health under the influence of the COVID pandemic that are covered by your study. As you change the title, change this part as well.
12. Line 301 is missing a reference for the claim made.
13. Line 372 is missing a full stop at the end of the sentence.
14. In line 432 there is a typographical error in the word Its.
15. References are not listed according to the rules for MDPI journals. Please correct all references as required by the rules of the journal.
After these corrections, I believe that the article will be accepted for publication.
Comments on the Quality of English LanguageModerate editing of English language is required, as far as I am concered.
Author Response
Dear Reviewer,
Firstly, we would like to express our gratitude for your insightful feedback and constructive criticism on our scientific paper. We appreciate your recognition of the originality and importance of our work, especially in the context of the Western world's response to the migrant health crisis during the COVID-19 pandemic.
We acknowledge your suggestions and have taken careful steps to address each point to improve the quality of our manuscript:
- Title Revision: We agree that the title should encompass more aspects of migrant life impacted by the pandemic. The title has been revised accordingly to reflect this.
- Typographical Error (Line 21): The extraneous letter 'A' has been removed.
- Keywords Adjustment: We have reorganized the keywords according to the MeSH system, ensuring they align more closely with the essence of our work. The term 'Transculturation' has been retired and replaced with more relevant terms.
- Literature Citations: Citations throughout the text have been corrected to follow standard rules, specifically in lines 56 and 71, as you pointed out.
- Study Type Clarification (Line 82): We have specified the type of study more clearly as 'descriptive, analytical, prospective, retrospective,' as per your suggestion.
- Expansion of Study Participants Section (Lines 97-100): We have provided clear inclusion and exclusion criteria for the examined group and elaborated on the reason for selecting only the male population, both cis and transgender, for the research.
- Theoretical Criteria for Sampling Reference (Line 122): An appropriate reference discussing the theoretical criteria for sampling has been added.
- Typographical Error (Line 136): The missing full stop at the end of the sentence has been inserted.
- Terminology Correction (Line 153): The term 'gay' has been reviewed and corrected to fit appropriately with the context.
- Results Section Enhancement: We have included an introductory graphic and textual sociodemographic and cultural presentation of the examined sample, addressing the lack of this information in the initial submission.
- Modification in Discussion Section (Line 298): As suggested, we have revised the discussion section to better encapsulate the various health aspects influenced by the COVID-19 pandemic, aligning it with the changes made to the title.
- Addition of Reference (Line 301): A reference has been included to support the claim made in this line, ensuring the accuracy and credibility of our assertions.
- Typographical Correction (Line 372): The missing full stop at the end of the sentence has been added.
- Correction of Typographical Error (Line 432): The typographical error in the word 'Its' has been corrected.
- References Formatting: We have revised all references to adhere to the rules for MDPI journals, ensuring they are correctly formatted as per the journal’s guidelines.
Reviewer 3 Report
Comments and Suggestions for Authors
Thank you for inviting me to examine the paper "Health of Migrant Men Living in Brazil During the Pandemic: A Qualitative Study."
The paper examines the influence of the SARS-CoV-2 pandemic on the daily lives of migrants, refugees, and asylum seekers, showing the challenges they faced from several aspects. The several health determinants are analyzed, but in a qualitative manner that limits generalization and comparison with other studies.
Overall, the paper is really fascinating and covers a variety of topics.
However, a section on Limitations is completely missing, as is research on the epidemiological characteristics of persons who have contracted COVID-19 among vulnerable groups. There are papers in Europe, but also in the Americas, that examine how migrant status leads to poorer access to health care and health inequality. I suggest including this part in the Introduction, emphasizing disparities in disease severity and other significant details.
In addition, when referring to persons who live in another country, the term "residing" is improper (Lines 23, 99). It is preferable to use the phrase "living" or one of its synonyms.
As per the methodology used, I am confident that further detail is required. It is in reality insufficient to describe Transcultural Care Theory without thoroughly defining what it is (Lines 83-85).
When referring to Excel (Line 128) in the same section, there is an inaccuracy: the application utilized by the Google Suite is "Google Sheets."
Line 142: "Country of origin" instead of "China."
As per the results, it would be beneficial to include a summary table at the end of the first section (after Line 158, to be clear). Despite the fact that this is a qualitative study, numerical data is available.
Finally, there is a section on Limitations that is completely lacking, as indicated above. The first question in this regard is: why did you just evaluate the male population? Is there any evidence to support your decision to exclude the female population? If so, explain it in the manuscript, in the Materials and Methods section, and note it as a limitation of the study, because we have no way of knowing if what was discovered is also applicable to women.
Furthermore, because the sample size is small, generalizations for the entire country of Brazil are not conceivable.
However, due to the nature of the study (qualitative), generalizations are not attainable, which is another significant weakness of this study.
Another limitation is the lack of a sample of subjects residing in Brazil as a control group: it would have been interesting to understand if these impairments were also shared by Brazilian individuals from various socioeconomic groups, and to what extent there were similarities and, at the same time, differences.
Author Response
Thank you once again for your detailed and constructive feedback on our manuscript. We have carefully reviewed your additional comments and have made the following revisions to enhance the quality and clarity of our paper:
- Modification in Discussion Section (Line 298): As suggested, we have revised the discussion section to better encapsulate the various health aspects influenced by the COVID-19 pandemic, aligning it with the changes made to the title.
- Addition of Reference (Line 301): A reference has been included to support the claim made in this line, ensuring the accuracy and credibility of our assertions.
- Typographical Correction (Line 372): The missing full stop at the end of the sentence has been added.
- Correction of Typographical Error (Line 432): The typographical error in the word 'Its' has been corrected.
- References Formatting: We have revised all references to adhere to the rules for MDPI journals, ensuring they are correctly formatted as per the journal’s guidelines.
Additionally, we have undertaken a moderate editing of the English language throughout the manuscript to enhance its readability and ensure it meets the high standards of academic writing.
- Correction in Line 142: The term "China" has been replaced with "Country of origin" to provide a more general and accurate description.
- Inclusion of Summary Table: Following your recommendation, we have added a summary table at the end of the first section (after Line 158). This table presents the numerical data available from our qualitative study, aiding in better comprehension of the results.
- Additional Context and Recommendations: We have expanded our discussion to include the effects of the pandemic on the daily lives of migrant men. This includes the impact on their mental health, social well-being, and the importance of cultural care, as well as the need for inclusive health policies and interventions.
- Explanation for Male Population Focus: In the Materials and Methods section, we have clarified our rationale for focusing solely on the male population. This decision was based on initial findings indicating specific challenges faced by this demographic in the pandemic context. We have also acknowledged this as a limitation of the study, noting the absence of female perspectives.
- Acknowledgement of Sample Size and Generalizability: We have included a statement acknowledging the small sample size and the consequent limitations it imposes on generalizing the findings to the entire country of Brazil.
- Limitations of Qualitative Nature: We have added a section discussing the inherent limitations of qualitative research, including the challenge of generalizing findings. This section further elaborates on the strengths and weaknesses of our methodological approach.
- Thank you for your insightful observation regarding the limitations of our study. We agree that the inclusion of a control group of subjects residing in Brazil would have provided a more comprehensive understanding of the impacts of the pandemic. This comparison could have highlighted similarities and differences between migrant men and Brazilian individuals from various socioeconomic groups.
In response to your feedback, we have included this aspect in our limitations section:
- The study lacks a control group of Brazilian residents, which limits the ability to compare and contrast the experiences of the migrant population with those of native Brazilians. This comparison could have offered insights into whether the impairments identified in our study are unique to the migrant population or shared by Brazilians from different socioeconomic backgrounds.
We have also emphasized the following points in the 'Limitations and Strengths' section of our paper:
The study identifies a vulnerable population with multiple transcultural needs, serving as an inducer of public policies supporting immigrants and highlighting the health needs of male immigrants during the pandemic.
However, it falls short in verifying behavioral changes among other nationalities due to time constraints or access to the research.
While the research reached participants from all regions of Brazil, the results may reflect specific realities, limiting generalizations. The study did not conduct a comparative analysis between men and women, limiting gender-based understandings of the investigated phenomenon.
The recruitment strategy may have also limited the research's reach among distinct groups of men. The cultural and territorial particularities of each region in Brazil might have resulted in specific experiences for each migrant man, hindering a deeper exploration of the findings.
Furthermore, we acknowledge the small sample size as a limitation, preventing generalizations for the entire country of Brazil. The inability to have a Brazilian control group due to the course of the pandemic and the changes imposed by immigration in this scenario is also noted.
We appreciate your thorough review and valuable feedback, which have significantly contributed to enhancing the overall quality of our study. We hope these revisions meet your expectations and look forward to the possibility of our article being accepted for publication.
Reviewer 4 Report
Comments and Suggestions for Authors
The article significantly contributes to the understanding of the multifaceted challenges faced by male immigrants during the COVID-19 pandemic in Brazil. The use of the Transcultural Nursing Theory as a framework for analysis offers a comprehensive lens to explore the cultural care implications arising from the pandemic.
The article as a whole is skillfully composed. The authors should edit the conclusion section and compose a paragraph instead of individual sentences.
Author Response
Dear Reviewer 4,
We are grateful for your positive assessment of our article and your acknowledgment of its significant contribution to understanding the challenges faced by male immigrants during the COVID-19 pandemic in Brazil. We also appreciate your recognition of the effective use of Transcultural Nursing Theory in our analysis.
In response to your suggestion regarding the conclusion section, we have revised it to present the information in a cohesive paragraph format instead of individual sentences. This adjustment enhances the readability and coherence of the conclusion, ensuring that it effectively summarizes our findings and their implications.
We believe this modification aligns with your guidance and improves the overall composition of the article. We are thankful for your constructive feedback, which has helped refine our manuscript. We look forward to the possibility of our revised article being accepted for publication.
Round 2
Reviewer 4 Report
Comments and Suggestions for Authors
The authors have made revisions to the manuscript based on my comments, and now the paper can be accepted in its current form.